# Characteristics of Solar Wind Radiation Damage in Lunar Soil: PAT and TEM Study

**DOI:** 10.3390/nano12071135

**Published:** 2022-03-29

**Authors:** Sizhe Zhao, Hongyi Chen, Yang Li, Shuoxue Jin, Yanxue Wu, Chuanjiao Zhou, Xiongyao Li, Hong Tang, Wen Yu, Zhipeng Xia

**Affiliations:** 1College of Earth Sciences, Guilin University of Technology, Guilin 541006, China; zhaosizhe@mail.gyig.ac.cn (S.Z.); chy@glut.edu.cn (H.C.); 2Center for Lunar and Planetary Sciences, Institute of Geochemistry, Chinese Academy of Sciences, Guiyang 550081, China; zhouchuanjiao@mail.gyig.ac.cn (C.Z.); lixiongyao@vip.skleg.cn (X.L.); tanghong@mail.gyig.ac.cn (H.T.); yuwen@mail.gyig.ac.cn (W.Y.); 3Center for Excellence in Comparative Planetology, Chinese Academy of Sciences, Hefei 230026, China; 4Multi-Disciplinary Research Division, Institute of High Energy Physics, Chinese Academy of Sciences, Beijing 100049, China; 5Analysis and Test Center, Guangdong University of Technology, Guangzhou 510006, China; wuyanxue1314@163.com; 6Key Laboratory of Planetary Geological Evolution at Universities of Guangxi Province, Guilin 541006, China; xiazhipeng@glut.edu.cn; 7Guangxi Key Laboratory of Hidden Metallic Ore Deposits Exploration, Guilin 541006, China

**Keywords:** positron annihilation, irradiation structural damage, vacancy defects

## Abstract

Irradiation structural damage (e.g., radiation tracks, amorphous layers, and vesicles) is widely observed in lunar soil grains. Previous experiments have revealed that irradiation damage is caused by the injection of solar wind and solar flare energetic particles. In this study, cordierite and gabbro were selected as analogs of shallow and deep excavated lunar crust materials for proton irradiation experiments. The fluence was 1.44 ± 0.03 × 10^18^ H^+^/cm^2^, which is equivalent to 10^2^ years of average solar wind proton implantation on the Moon. Before and after irradiation, structural damage in samples is detected by slow positron annihilation technology (PAT), Doppler broadening (DB) measurement, focused ion beam (FIB), and transmission electron microscopy (TEM). The DB results showed the structural damage peaks of irradiated gabbro and cordierite were located at 40 and 45 nm. Hydrogen diffused to a deeper region and it reached beyond depths of 150 and 136 nm for gabbro and cordierite, respectively. Hydrogen atoms occupied the original vacancy defects and formed vacancy sites—hydrogen atom complexes, which affected the annihilation of positrons with electrons in the vacancy defects. All of the DB results were validated by TEM. This study proves that the positron annihilation technique has an excellent performance in the detection of defects in the whole structure of the sample. In combination with TEM and other detection methods, this technology could be used for the detection of structural damage in extraterrestrial samples.

## 1. Introduction

Without protection from a thick atmosphere, lunar surface materials are directly exposed to solar wind particles. Previous studies of lunar soil samples collected by the Apollo and Luna missions have proved that micro/nanostructure damage, including lattice defects, amorphous rims, and vesicles, are widely distributed on the top surface layer of mature lunar soil particles [1,2,3]. The density of radiation damage tracks and the degree of amorphization of various types of minerals can be used to evaluate the exposure age of lunar soil grains. In addition, the vesicles produced by saturation injection and gathering of solar wind gas ions (mainly H^+^ and He^+^) are ideal reservoirs for OH^−^ and ^3^He, which are believed to be potential sources of water and nuclear fusion materials on the Moon [4]. Therefore, the irradiation damage of lunar mineral grains plays an important role in studying lunar soil’s evolution history and resource utilization. 

The Chang’E-5 mission successfully returned 1731 g of lunar soil (~1500 g obtained by shoveling and ~231 g obtained by drilling) from the Mairan domes, northeast of the Procellarum KREEP Terrane. The eruption age (~2 billion years) of basalt (Em4/P58) and ejection age (~5 to 100 Ma) of the lunar regolith at the landing site of Chang’E-5 were measured and calculated [5,6]. The results showed that the exposure age of the Chang’E-5 lunar soil was much shorter than those of the Apollo and Luna soil samples. Therefore, the Chang’E-5 lunar sample provides a new perspective from which to study the evolutionary history of lunar soil. The key questions include but are not limited to, the precise exposure ages of the shoveled and drilled samples and their relation to the irradiation structural damage degree caused by solar wind and solar flare particles; the microstructure damage characteristics of the soil grains through the cross-section of the lunar soil at the landing site revealed by the drilled samples.

Previous studies of the irradiation damage have mainly depended on microanalysis techniques, especially scanning electron microscopy (SEM) and TEM [7,8,9]. However, it is unrealistic to obtain a complete understanding of lunar soil’s irradiation structural damage degree through microanalysis of a single or dozens of lunar soil particles at the nanometer scale. The positron annihilation technique enables macroscopic characterization of structural damage without damage or contamination of the sample [10] and it has been widely used in the field of nuclear materials. The main technologies that are commonly used in PAT are positron annihilation lifetime spectroscopy (PALS), DB, and angular correlation annihilation radiation (ACAR). In the above technologies, DB has the optimal suitability for surface structure detection. There are two main ways for positron annihilation: annihilation with electrons with the generation of a pair of photons and annihilation through capture by the structural defect. These techniques characterize positron annihilation in different ways to obtain information about the structural damage of the target sample.

In this study, we introduced DB into the analysis of the irradiation damage of lunar soil samples. We chose gabbro, mainly composed of pyroxene and plagioclase, and cordierite as the representative minerals of shallow and deep excavated lunar crust materials. First, all of the samples were mounted and implanted by a proton ion implanter to simulate the implantation of solar wind protons. Second, the irradiated and unirradiated samples were measured by slow positron annihilation DB measurement to obtain the degrees, distributions, and positions of radiation damage. Third, to verify the analysis results of DB, FIB-TEM was used to obtain the microstructure characteristics of the target minerals.

## 2. Materials and Methods

### 2.1. Sample

Gabbro and cordierite were selected as lunar material analogs. The pyroxene and plagioclase in gabbro are common mineral phases on the Moon, which form at a depth of ~20 km [11]. Previous studies based on the samples returned by Apollo 15 have found that cordierite-spinel troctolite is present in the lunar highlands [12]. It was suggested that it formed 10 km above the lunar crust-mantle boundary and was excavated by large impact events. Gabbro and cordierite represent shallow and deep excavated lunar crust materials, respectively. In addition, cordierite is related to common lunar minerals, for example, it has the same crystal structure as olivine (orthorhombic crystal system) and similar chemical composition (Mg_2_Al_4_Si_5_O_18_) to anorthite (CaAl_2_Si_2_O_8_). Analysis of gabbro and cordierite helps to understand the space weathering characteristics of deep lunar excavation materials.

Cordierite was collected in India and contained tiny zircon (Figure 1a). Gabbro was collected in Tibet, China. The main mineral phases of the gabbro sample were pyroxene and plagioclase (Figure 1b). Both cordierite and gabbro were processed into thin sections (12 mm × 12 mm × 3 mm) with a diamond wire saw and their surfaces were polished with silicon carbide abrasive papers (#400, #800, and #2000). The cordierite and gabbro samples were then ultrasonically cleaned with alcohol and acetone for 20 min, respectively. The samples were finally oven-dried at 120 °C for 12 h to remove the remaining moisture from the samples.

### 2.2. Hydrogen Ion Irradiation

The experiments were performed with the high vacuum ion implanter system (CNTT Co., Ltd. Chengdu, SC, China) at the Institute of Geochemistry, Chinese Academy of Science (CAS), Guiyang. This system contained an ion source, a high-vacuum system (maximum vacuum of 6 × 10^−5^ Pa), a magnetic deflection field, and an analyzer. The studied samples were mounted on a specially made transferable sample holder (irradiated area of ~6 cm^2^), which can measure the ion-beam flux at the sample surface. H^+^ was chosen as the irradiation source because H^+^ is the main component of the solar wind (~95%) and irradiation damage is mainly caused by H^+^ [13]. Under the vacuum of 10^−5^ Pa, H^+^ passed through the magnetic deflection field and was implanted into the samples with an energy of 1.5 keV. This energy is within the range of the solar wind energy (0.3–3 keV/u) on the lunar surface [13]. To ensure that structural damage could be generated in the samples, the total irradiation time was ~20 h and the fluence was 1.44 ± 0.03 × 10^18^ H^+^/cm^2^, which is equivalent to 10^2^ years of average solar wind proton implantation on the Moon [14].

### 2.3. PAT

DB measurement was carried out at the slow positron beam facility of the Institute of High Energy Physics, CAS, Beijing. The system consisted of a positron source (^22^Na), an ultrahigh vacuum system (maximum vacuum of 3 × 10^−7^ Pa), and a coincidence DB spectrometer (Figure 2). The DB spectrometer detected the γ-photons produced by positron annihilation through high-purity germanium detectors. The total peak energy range of the collected γ-energy spectrum was 499.5–522.5 keV. The positron beam energy range was set from 0.18 to 10 keV.

The detection depth of the slow positron is defined by the incident energy and it is calculated by the empirical equation [15]
(1)$$R=\left(\frac{40}{\rho }\right)E^{1.6}\;$$
where *R* is the slow positron detection depth (nm), *ρ* is the density of the sample (g/cm^3^), and *E* is the slow positron incident energy (keV). According to Equation (1), the slow positron detection depth is shown on the top x-axis of the *S*–*E* curves. The maximum detection depth was >500 nm, which covered the damage profiles and distribution of hydrogen ions.

The DB results are mainly characterized by the *S* parameter, *W* parameter, and Δ*S*. The *S* parameter is defined as the ratio between the counts in the energy range 510.2–511.8 keV (low momentum electron area) and the total peak (499.5–522.5 keV) counts, which can be used for characterizing the structural damage. A larger *S* parameter indicates more structural damage. The *W* parameter is defined as the ratio between the counts in the energy ranges 513.6–516.9 keV and 505.1–508.4 keV (high momentum electron area) and the total peak (499.5–522.5 keV) counts. Combined with the *S* parameter, the slope of the *S*–*W* plot can be used to characterize the mechanism of positron annihilation. Δ*S* is calculated by the equation [16]
(2)$$\Delta S=S_{irradiated}\;–\;S_{unirradiated}$$
where *S_irradiated_* is the *S* parameter after irradiation and *S_unirradiated_* is the *S* parameter before irradiation. Δ*S* indicates the structural damage only induced by hydrogen ion irradiation.

### 2.4. FIB-TEM

Before irradiation, back-scattered electron (BSE) mapping was performed at 15 keV using the FEI Scios (FEI Inc., Hillsboro, OR, USA) Dual-beam system at the Institute of Geochemistry, CAS, Guiyang. This system combines SEM, FIB, and energy dispersive spectroscopy (EDS). After DB measurement, the ultrathin cross-section for TEM was prepared by FIB at 30 kV/3 nA for digging and 5 kV/48 pA and 2 kV/43 pA for polishing. The final thickness was 100 nm. The FIB cross-sections were characterized using the FEI Talos F200S electron microscope (FEI Inc., Hillsboro, OR, USA) at the Guangdong University of Technology. High-resolution (HR) TEM images and selected area electron diffraction (SAED) patterns were obtained to confirm the nanoscale structure.

## 3. Results and Discussion

After irradiation, the surfaces of the samples did not significantly change. SEM secondary electron images of the irradiated gabbro and cordierite samples are shown in Figure 3d. The surface morphological characteristics essentially preserved the initial shapes. The same slight grooves shown in Figure 3d were also observed before irradiation. The grooves were caused by sample polishing and not by irradiation. This provided direct evidence that hydrogen ion irradiation at 1.5 keV and 1.44 ± 0.03 × 10^18^ H^+^/cm^2^ hardly affected the surface character. To verify the analysis results of DB, we selected three representative FIB sites in cordierite, plagioclase, and pyroxene, as shown in Figure 3a–c.

The *S* (*S* parameter)–*E* (positron energy) curves and TEM bright-field micrographs of gabbro are shown in Figure 4. The structural damage peak of irradiated gabbro was located at 40 nm (Figure 4a). According to the TEM results in Figure 4b,c, the centers of the plagioclase and pyroxene vesicles were mostly at 40 nm. The pyroxene vesicles were large (average diameter of ~20 nm, maximum diameter of ~200 nm) but less in number, while those of plagioclase were the opposite (small but more in number, average diameter of ~20 nm, maximum diameter of ~90 nm). The DB results showed that *S_irradiated_* < *S_unirradiated_* were in the depth region >150 nm. Combined with the TEM results, channel effects occurred in this region. The channel effects in gabbro are shown in Figure 4c,e. Directionally arranged channels with different lengths and widths formed below the vesicles. The channels formed in pyroxene and plagioclase were quite different. Plagioclase had longer and wider channels than pyroxene. Lengths were from tens to hundreds of nanometers with a width of approximately 5 nm. Pyroxene had finer and more channels than plagioclase, with lengths ~100 nm and widths of no more than 1 nm. These channels may become the medium for hydrogen diffusion in gabbro. The results of previous studies have shown that a small amount of hydrogen diffuses to the deep region through the channels and is deposited at the original structural damage. Hydrogen atoms occupy the vacancy sites and form “vacancy sites–hydrogen atom complexes” [17,18,19]. The formation of complexes affects the annihilation of positrons with electrons in the vacancy defects, leading to a reduction of the *S* parameter in the depth region >150 nm. In addition to verifying the DB results, TEM provided further information about the structural damage of the sample. The average amorphous layer thickness was 300 nm in plagioclase and 400 nm in pyroxene. However, irradiation did not make the sample completely amorphous. Some clear lattice fringes were observed in the plagioclase amorphous layer (Figure 4d), the same phenomenon was found in previous work [20].

The *S*–*E* curves and TEM bright-field micrographs of cordierite are shown in Figure 5. The unirradiated cordierite sample showed an increasing trend of *S* parameters with increasing positron detection depth in the range of 136–590 nm. The TEM results revealed the reasons for this trend (Figure 5b). In the 136–590 nm depth region, multiple original dislocations were observed, leading to an increase in *S* parameters. The *S*–*E* curve showed that the irradiation structural damage peak was located at 45 nm. An HRTEM image of the vesicles is shown in Figure 5c. The depth of 45 nm marked by the white dashed line is the depth at which the vesicles were mainly concentrated. These vesicles were relatively uniform in size (mostly around 20 nm in diameter) with only a few large vesicles among them, but their diameter did not exceed 90 nm.

The TEM results showed other details about irradiation modification. The average thickness of the amorphous layer was approximately 85 nm. We only observed one channel effect in the sample (width of <1 nm, length of <100 nm, Figure 5c), and no stress streaks were observed. The characteristics of the three FIB cross-sections are compared in Table 1.

Among the three samples, irradiation produced the least structural damage in cordierite (Table 1), which may be caused by the different ion connections in the minerals. Pyroxene, plagioclase, and cordierite belong to the groups of inosilicates, tectosilicates, and cyclosilicates, where the compactness order is cyclosilicates > inosilicates > tectosilicates [21]. Due to the minerals’ different compactness, irradiation produces the most structural damage in plagioclase, followed by pyroxene, and the least structural damage is produced in cordierite.

To verify the structural damage triggered by proton irradiation, the Δ*S* curves and *S*–*W* plots of the samples are shown in Figure 6. Δ*S* of cordierite showed a slightly decreasing trend in the depth range of 136–590 nm (Figure 6a). As previously mentioned, a large number of original dislocations at this depth range lead to a clear increasing trend of *S* parameters. The opposite trend for Δ*S* indicates that even though irradiation produced very small channels in cordierite, a small amount of hydrogen could still diffuse along the channels and occupy the original dislocations of the sample. Δ*S* of cordierite was higher than that of gabbro, which slightly disagrees with the TEM results. Cordierite is a cyclosilicate, and irradiation does not easily cause serious structural damage to cordierite. The vesicles produced in cordierite were smaller than those produced in gabbro, but they may be larger in number. Since we did not produce more FIB cross-sections in this study, we cannot conclude the exact number of vesicles. Cordierite may also contain atomic-level structural damage, which can also cause an increase in the *S* parameter, but we were unable to verify this owing to TEM resolution limitations. PAT is statistically superior to FIB-TEM because the selection of the FIB cross-section location is somewhat random. The locations that we selected could differ from the general trend and have less structural damage, which could also be the reason for the slight disagreement between the DB and TEM results.

Previous studies have shown that the *S* and *W* parameters are generally inversely related [22,23,24]. The *S*–*W* plots of the samples are shown in Figure 6b. The (*S*, *W*) points of both gabbro and cordierite showed a tendency to move to the lower right after irradiation, which indicates that irradiation increased the size of the structural damage on the samples. However, the (*S*, *W*) points of cordierite were closer to the lower right both before and after irradiation. As previously discussed, cordierite contained multiple original dislocations in the 136–590 nm depth range (Figure 5b), resulting in a higher *S* parameter than gabbro. Owing to the dense structure of cordierite, hydrogen diffusion in cordierite is weak, which leads to its *S* parameters remaining higher than that of gabbro after irradiation. The *W* parameters of both gabbro and cordierite showed the trend of increasing and then decreasing as the positron penetration depth increased after irradiation. This trend, which is consistent with the TEM results, suggests that the annihilation mechanisms of positrons after irradiation were not the same in the near-surface and deeper regions (Figure 4c,d and Figure 5b). The different positron annihilation mechanisms are indicated with solid and dashed ellipses in Figure 6b.

## 4. Conclusions

The structural damage characteristics of hydrogen irradiated gabbro and cordierite were investigated by DB and FIB-TEM. After irradiation, the structural damage peaks of gabbro and cordierite were located at 40 and 45 nm, respectively. Irradiation produced different numbers and sizes of channels. Hydrogen diffused through the channels in the samples. Below depths of 150 and 136 nm in gabbro and cordierite, respectively, hydrogen deposited at the original structural damage occupied the vacancy sites and formed vacancy sites—hydrogen atom complexes. The experimental results showed that PAT is a useful technique for structural detection because it can accurately obtain information about the structural damage in the surface layer of the sample. In combination with TEM results, comprehensive information about structural damage types, distribution, and sizes can be obtained. This technology could be used in the future to analyze extraterrestrial samples for rapid analysis of structural damage.

## Figures and Tables

**Figure 1 nanomaterials-12-01135-f001:**
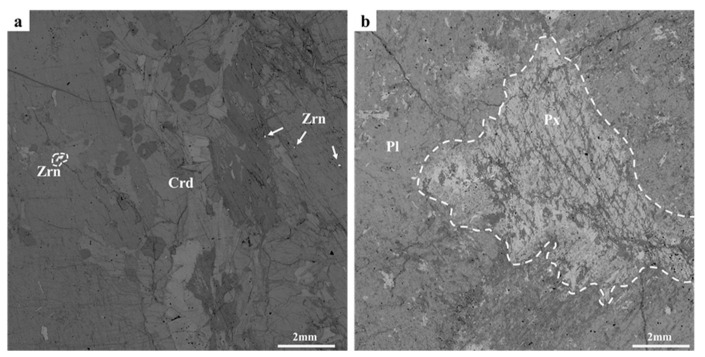
BSE images of (**a**) cordierite and (**b**) gabbro. Cordierite mainly consisted of panel/grained cordierite (~98 vol.%) with a small amount of grained zircon (~2 vol.%). Gabbro was characterized by the gabbro texture, and it was mainly composed of pyroxene vein (~30 vol.%) and plagioclase phenocryst (~70 vol.%). Crd, cordierite; Zrn, zircon; Pl, plagioclase; Px, pyroxene.

**Figure 2 nanomaterials-12-01135-f002:**
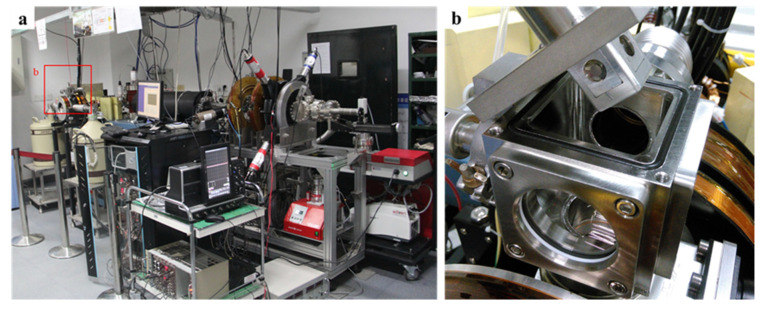
(**a**) Slow positron annihilation DB facility. (**b**) Sample assembly.

**Figure 3 nanomaterials-12-01135-f003:**
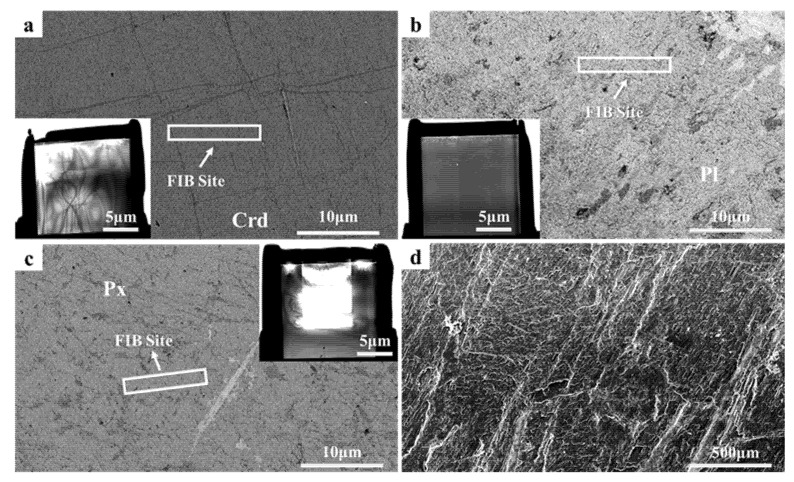
SEM back-scattered electron images of (**a**) cordierite, (**b**) plagioclase, and (**c**) pyroxene. The red rectangles indicate the extraction locations of the FIB samples. The inserts in the bottom left of (**a**,**b**) and top right of (**c**) show scanning transmission electron microscope images of the FIB cross-section. (**d**) Secondary electronic image of the gabbro surface after irradiation.

**Figure 4 nanomaterials-12-01135-f004:**
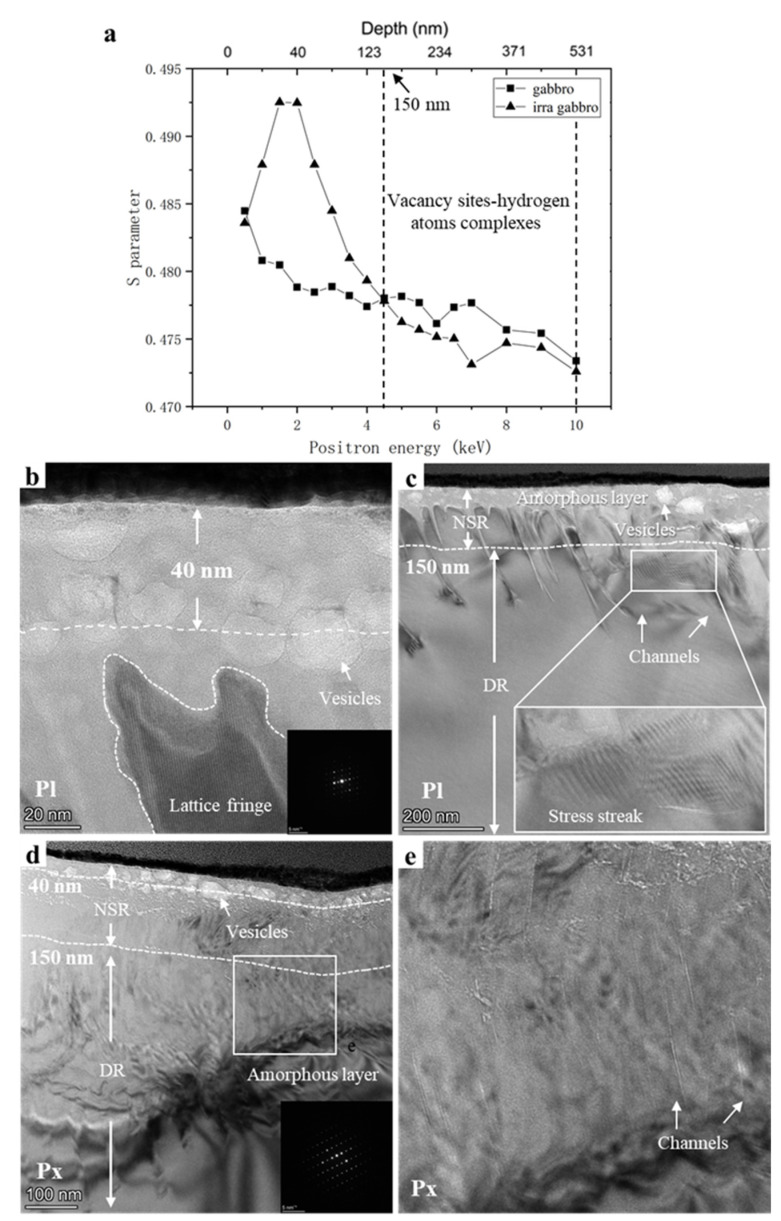
(**a**) *S*–*E* curves of gabbro and irradiated gabbro. The white dashed lines indicate the vacancy sites—hydrogen atom complex region. (**b**) TEM bright-field and (**c**) HRTEM images of plagioclase. The SAED pattern of plagioclase is shown in the insert in the bottom right corner of (**b**). The white dashed line represents the current depth. (**d**) TEM bright-field and HRTEM images of pyroxene. (**e**) HRTEM images of the channel effect in Figure d. The SAED pattern of plagioclase is shown in the insert in the bottom right corner of (**d**). The white dashed line represents the current depth. NSR, near-surface region; DR, deeper region.

**Figure 5 nanomaterials-12-01135-f005:**
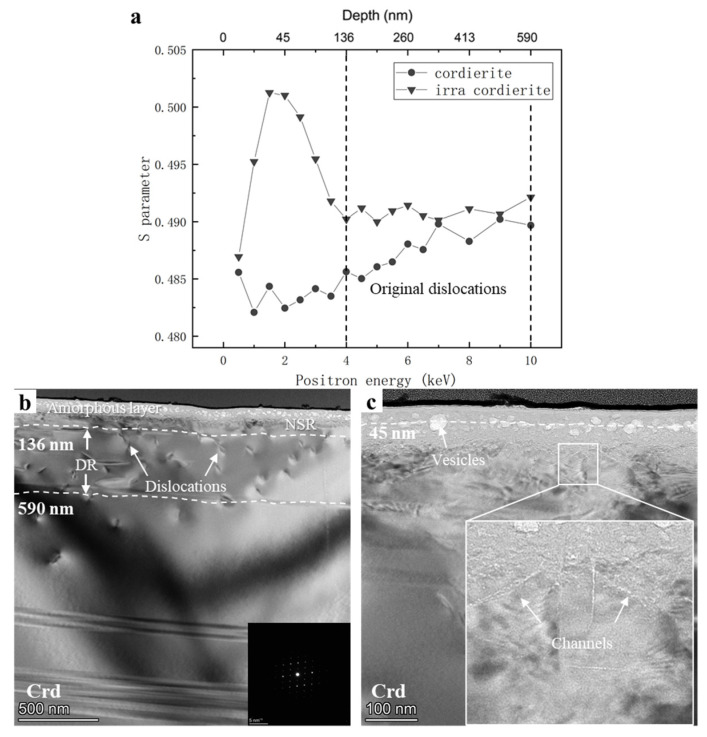
(**a**) *S*–*E* curves of cordierite and irradiated cordierite. The black dashed line represents the original dislocation region. (**b**) TEM bright-field and (**c**) HRTEM images of plagioclase. The SAED pattern of cordierite is shown in the insert in the bottom right corner of (**b**). The white dashed lines represent the current depth. NSR, near-surface region; DR, deeper region.

**Figure 6 nanomaterials-12-01135-f006:**
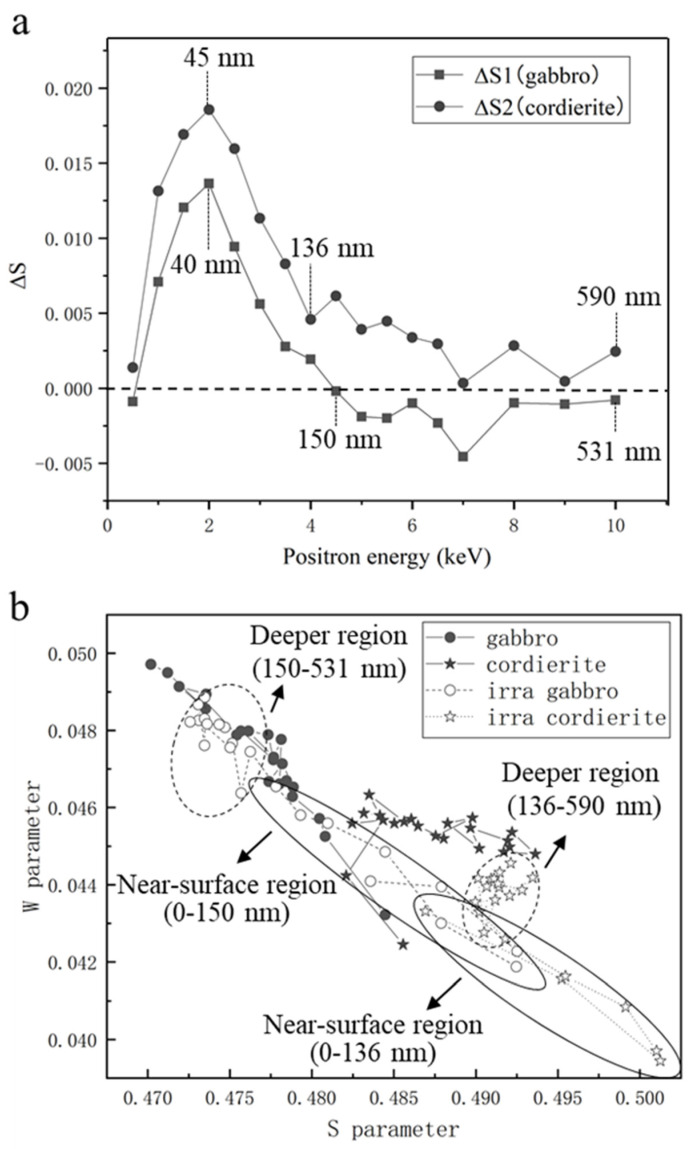
(**a**) Δ*S* curves of gabbro and cordierite. The black vertical dashed lines represent the current point detection depths of gabbro and cordierite, respectively. The horizontal dashed black line is the baseline (Δ*S* = 0). (**b**) *S*–*W* plots of unirradiated and irradiated gabbro and cordierite, which show the different mechanisms of positron annihilation in the near-surface and deeper regions. The solid and dashed ellipses represent the near-surface and deeper regions, respectively.

**Table 1 nanomaterials-12-01135-t001:** Modification characteristics of the three FIB cross-sections.

	Amorphous Layer	Vesicles	Channels Effect	Stress Stripe
**Plagioclase**	Avg. ~300 nm	Avg. ~20 nm, Max. ~200 nm	Multiple Length < 600 nm Width < 5 nm	Yes, multiple
**Pyroxene**	Avg. ~400 nm	Avg. ~20 nm, Max. ~90 nm	Less Length < 100 nm Width < 1 nm	Yes, less
**Cordierite**	Avg. ~85 nm	Avg. ~20 nm, Max. ~90 nm	Tiny Length < 100 nm Width < 1 nm	No

## Data Availability

Data are contained within the article.

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
