# Peer review of "Characteristics of Solar Wind Radiation Damage in Lunar Soil: PAT and TEM Study"

_nanomaterials, 2022, doi:10.3390/nano12071135_

Round 1
Reviewer 1 Report
The manuscript is well written, and the methodologies are well explained and detailed. Conclusions are supported by the data and properly interpreted. Because of these I recommend this work to be published with a couple of minor revisions.
Doppler Broadening abbreviation (DB) appears in the introduction directly without previous definition. Even when DB was presented in the abstract, the rest of the abbreviations, such as TEM, are again introduced in the text even when they were present in the abstract. I suggest doing the same for DB.
Figure 6 a. I suggest adding the top and right lines of the plot are to keep the same aesthetics as the same graphs on the work.
Author Response
Dear reviewers,
We have carefully considered your suggestion and made some changes, here are the responses to your comments:
"Doppler Broadening abbreviation (DB) appears in the introduction directly without previous definition. Even when DB was presented in the abstract, the rest of the abbreviations, such as TEM, are again introduced in the text even when they were present in the abstract. I suggest doing the same for DB."
Response: We agree with you that DB should appear in the introduction with the previous definition, however, in line 26, we have explained the meaning of DB. Therefore, we suggest that this meaning not be reintroduced in the introduction.
"Figure 6 a. I suggest adding the top and right lines of the plot are to keep the same aesthetics as the same graphs on the work."
Response: That is a very good suggestion, and we have already added the top and right lines of the plot in Figure 6 a.
Reviewer 2 Report
Good paper, Good technical English.
Research conclusions are supported by original measurements.
Proper combining of PAT, SEM and TEM methods.
Reference list could be improved and extended mainly in chapter 3 (Results and discussion).
Author Response
Dear reviewers,
Thank you very much for your careful review of our manuscript. We have tried our best to improve and made some changes in the manuscript, here are the responses to your comments:
"Reference list could be improved and extended mainly in chapter 3 (Results and discussion)."
Response: We are very sorry for the insufficient references provided and it is rectified at Line 199 and 264.